# Reproduction of GANSpace

## Reproducibility Summary

**Scope of Reproducibility**

The authors introduce a novel approach to analyze Generative Adversarial Networks (GANs) and create interpretable controls for image manipulation and synthesis. This is done by identifying important latent directions based on Principal Component Analysis (PCA) applied either in the latent space or the feature space. We aim to validate the claims and reproduce the results in the original paper.

**Methodology**

The code that was provided by the authors in Pytorch was reimplemented in **Tensorflow 1.x** for the pretrained *StyleGAN* and *StyleGAN2* architectures. This was done with the help of the APIs provided by the original authors of these models. The experiments were run on an Intel i7 processor containing 16 GB of RAM, coupled with an Nvidia 1060 GPU having 6 GB of VRAM.

**Results**

We were able to reproduce the results and verify the claims made by the authors for the StyleGAN and StyleGAN2 models by recreating the modified images, given the seed and other configuration parameters. Additionally, we also perform our own experiments to identify new edits and show that edits are transferable across similar datasets using the techniques proposed by the authors.

**What was easy**

The paper provides detailed explanations for the different mathematical concepts that were involved in the proposed method. This, augmented with a well-structured and documented code repository, allowed us to understand the major ideas in a relatively short period of time. Running the experiments using the original codebase was straightforward and highly efficient as well, as the authors have taken additional steps to employ batch processing wherever possible.

**What was difficult**

Originally we were attempting to recreate identical images with zero delta in the RGB values. However, due to differences in the random number generators between PyTorch-CPU, PyTorch-GPU and Numpy, the random values were not the same even with the same seed. This resulted in minute differences in the background artifacts of the generated images. Additionally, there is a lack of open source Tensorflow 1.x APIs to access the intermediate layers of the *BigGAN* model. Due to time constraints, we were unable to implement these accessors and verify the images that the authors of GANSpace created using *BigGAN*.

**Communication with original authors**

While conducting our experiments, we did not contact the original authors. The paper and codebase were organized well and aided us in effectively reproducing and validating the authors' claims.

# 1  Introduction

Generative Adversarial Networks (GANs) [1] are a type of machine learning framework where two neural networks, the discriminator and the generator, compete with each other in a zero-sum game. The generator tries to trick the discriminator into believing that artificially generated samples belong to real data.

GANs have proven to be powerful image synthesis tools, which are capable of producing high quality images. However, they provide little control over the features of the generated image. Existing solutions to add user control over the generated images require expensive supervised training on latent vectors.

GANSpace [2] proposes a simple technique to discover interpretable GAN controls in a unsupervised manner. The authors show that important directions in the latent space that affect the output can be identified using Principal Component Analysis (PCA). Their experiments on StyleGAN [3], StyleGAN2 [4] and BigGAN512-deep [5] demonstrate that layer-wise decomposition of PCA directions leads to many interpretable controls, which affects both low and high level attributes of the output image.

# 2  Scope of reproducibility

For our reproduction study, we aim to validate the effectiveness of the proposed technique in offering powerful interpretable controls on the output images in an unsupervised manner.

The following claims of the paper have been verified and tested successfully:

- PCA can be used to highlight important directions in the GAN's latent space.

- The GAN's output can be controlled easily in an unsupervised fashion.

- The earlier components control the higher-level aspects of an image, while the later directions primarily affect the minute details.

# 3  Methodology

The output of StyleGAN and StyleGAN2 can be controlled by identifying principal axes of $p(\mathbf{w})$, which is the probability distribution of the output of the mapping network $M$. First, we sample $N$ latent vectors $\mathbf{z}_{1:N}$ and compute the corresponding $\mathbf{w}_i = M(\mathbf{z}_i)$. The PCA of these $\mathbf{w}_{1:N}$ values gives us the basis $\mathbf{V}$ for $\mathcal{W}$. The output attributes of a new image given by $\mathbf{w}$ can then be controlled by varying the PCA coordinates of $\mathbf{x}$ before feeding them into the synthesis network.

$$\mathbf{w'} = \mathbf{w} + \mathbf{V}\mathbf{x} \tag{1}$$

Each entry $x_k$ of $\mathbf{x}$ is a separate control parameter which can be modified to update the desired attributes of the output image.

We follow the same notation used by the authors to denote edit directions in this report. $E(\mathbf{v}_i, j - k)$ means moving along component $v_i$ from layers $j$ to $k$.

## 3.1  Model descriptions

We use NVIDIA's official implementation of StyleGAN [1] and StyleGAN2 [2] models. The authors code for computing PCA on the latent space of StyleGAN was modified to support the API's provided by NVIDIA.

## 3.2  Datasets

The experiments in the paper were performed using the FFHQ, LSUN Car, CelebAHQ, Wikiart, Horse and Cat datasets. The official Tensorflow implementation of StyleGAN contains links to download pretrained models on FFHQ, LSUN Car, Wikiart, Horse and Cat. The models trained on Wikiart were downloaded from awesome-pretrained StyleGAN [3].

---

[1] https://github.com/NVlabs/stylegan
[2] https://github.com/NVlabs/stylegan2
[3] https://github.com/justinpinkney/awesome-pretrained-stylegan

69  In addition to the datasets using by the authors, we also perform our own experiments on the Beetles and Anime datasets
70  which were downloaded from awesome-pretrained StyleGAN2 [4].

## 3.3 Experimental setup

72  All the experiments were conducted on a laptop with an Intel i7 8750H processor, 16GB RAM, NVIDIA GTX 1060 6
73  GB GPU and Ubuntu 18.04. The generated images from our experiments were evaluated visually to determine whether
74  the edits were working as expected.

# 4 Results

76  First we validate the claims of the orignal paper mentioned in section 2. Then we move on to provide additional results
77  that validate the effectiveness of the technique employed by GANSpace.

## 4.1 Effectiveness of PCA

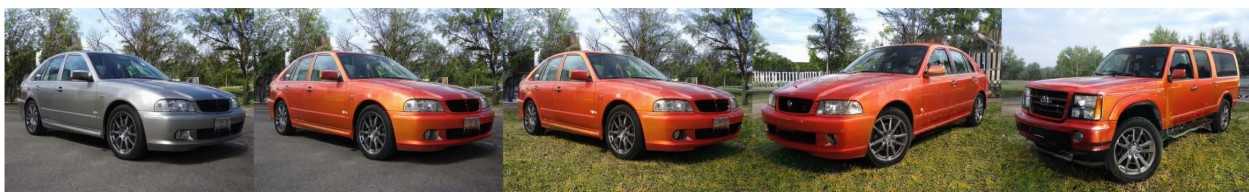

Figure 1: Sequences of image edits performed using control discovered with StyleGAN2 cars: "Initial Image" →
"Change Color" → "Add Grass" → "Rotate" → "Change Type"

79  Figure 1 highlights the effectiveness of PCA on changing low and high level attributes of the image. We are able to
80  control object shape, colour and pose as well as nuanced landscape attributes.

81  The edit directions corresponding to each of the edits are: $E(\mathbf{v}_{22}, 9-10)$ ("Change Color"), $E(\mathbf{v}_{11}, 9-10)$ ("Add
82  Grass"), $E(\mathbf{v}_0, 0-4)$ ("Rotate") and $E(\mathbf{v}_{16}, 3-5)$ ("Change type").

---

[4] https://github.com/justinpinkney/awesome-pretrained-stylegan2

## 4.2 Unsupervised vs Supervised methods

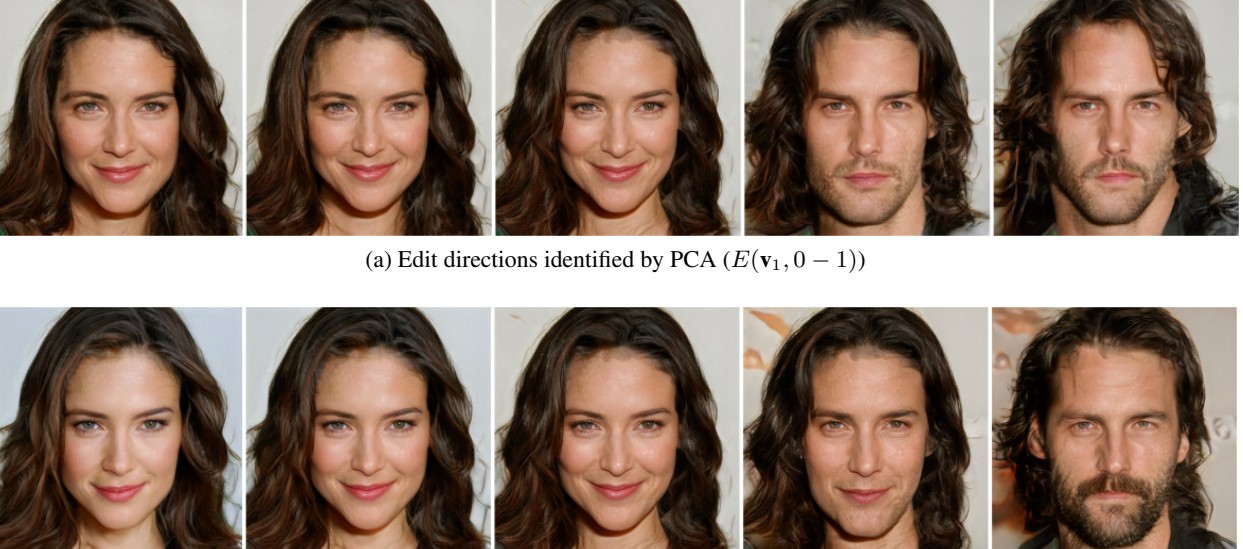

(a) Edit directions identified by PCA ($E(\mathbf{v}_1, 0-1)$)

(b) Edit directions identified by supervised methods [6]

Figure 2: Comparison of edits using unsupervised and supervised methods

The original authors point out that previous methods for finding interpretable directions in GAN latent spaces require outside supervision, such as labeled training images or pretrained classifiers, whereas GANSpace aims to automatically identify variations intrinsic to the model without supervision. This has been validated using the CelebA HQ Faces dataset by comparing the edit directions found through PCA to those found in previous work using supervised methods.

## 4.3 Effect of different components

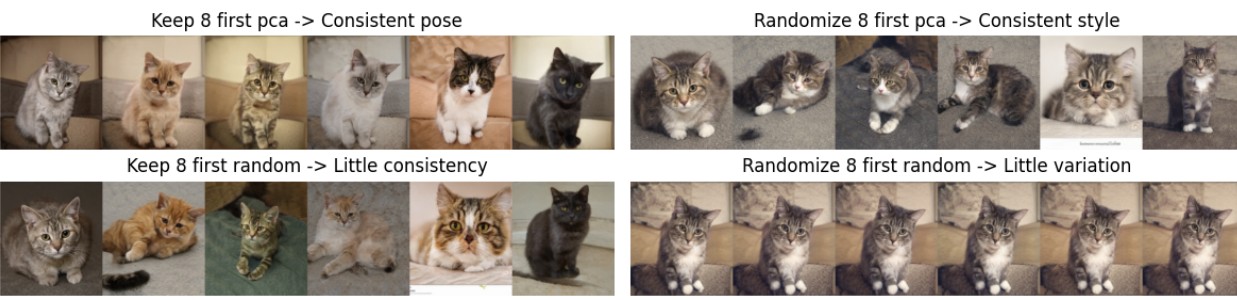

Figure 3: Illustration of the significance of the principal components as compared to random directions in the intermediate latent space of StyleGAN2.

The original authors claim that the earlier components primarily control the geometry and other high-level aspects, while the lower components capture minute details. This has been illustrated in 3. Fixing and randomizing the early principal components shows a separation between pose and style. In contrast, fixing and randomizing randomly-chosen directions does not yield a similar meaningful decomposition.

## 4.4 Additional results not present in the original paper

### 4.4.1 New edits

We identify new edits on the Stylegan2 Beetles dataset. Edit $E(\mathbf{v}_2, 0-17)$, referred to as "Patterns", adds a pattern on the shell of the beetle. The generated pattern varies depending on the seed used to sample $\mathbf{w}$.

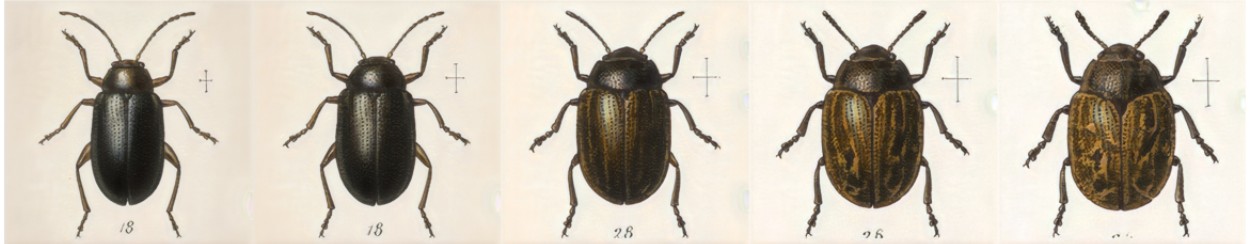

(a) Beetle generated with seed 1819967864

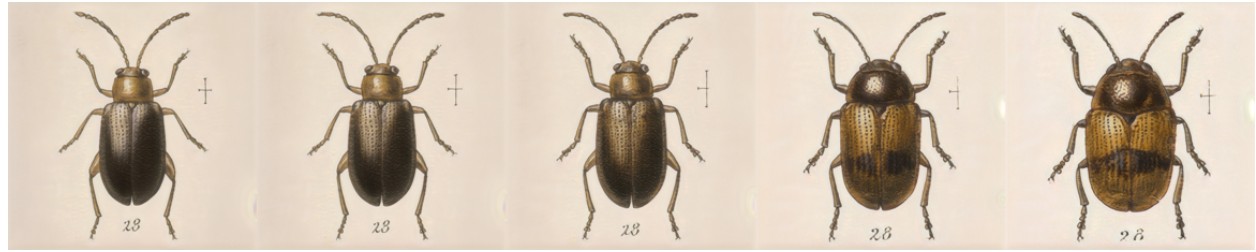

(b) Beetle generated with seed 1

Figure 4: "Patterns" edit applied on the output images of StyleGAN2 Beetles

### 4.4.2 Transferable edits across similar datasets

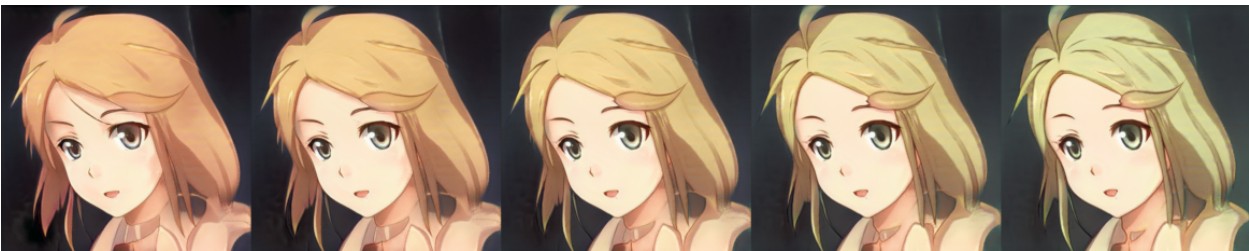

(a) Hair Color generated with seed 452352 on the Anime Portraits dataset



(b) Hair Color generated with seed 452352 on the FFHQ dataset

Figure 5: "Hair Color" edit applied on the output images of StyleGAN2 Anime Portraits and StyleGAN2 FFHQ datasets

The original authors limit the application of edits to the same dataset. We additionally show that the edits are transferable across datasets, provided that the seed values generate similar images. This has been illustrated in 5.

### 4.4.3 Truncation Psi on StyleGAN

The original authors use the "truncation trick" on images generated using StyleGAN2 to improve their quality. However, this is not enabled for StyleGAN images. During our experimentation, we found that enabling truncation while applying edits on StyleGAN images improved their quality as well. We demonstrate this using the Wikiart dataset using the "Head Rotation" ($E(\mathbf{v}_7, 0 - 1)$) and "Simple Strokes" ($E(\mathbf{v}_9, 8 - 14)$) edits.

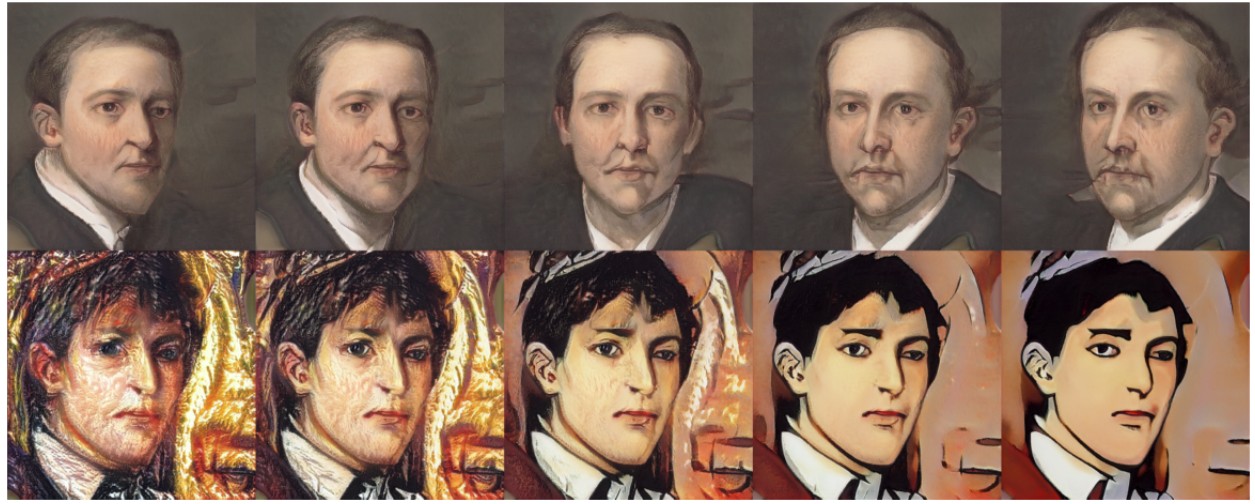

(a) "Head Rotation" and "Simple Strokes" edits on StyleGAN Wikiart with truncation psi set to 0.7

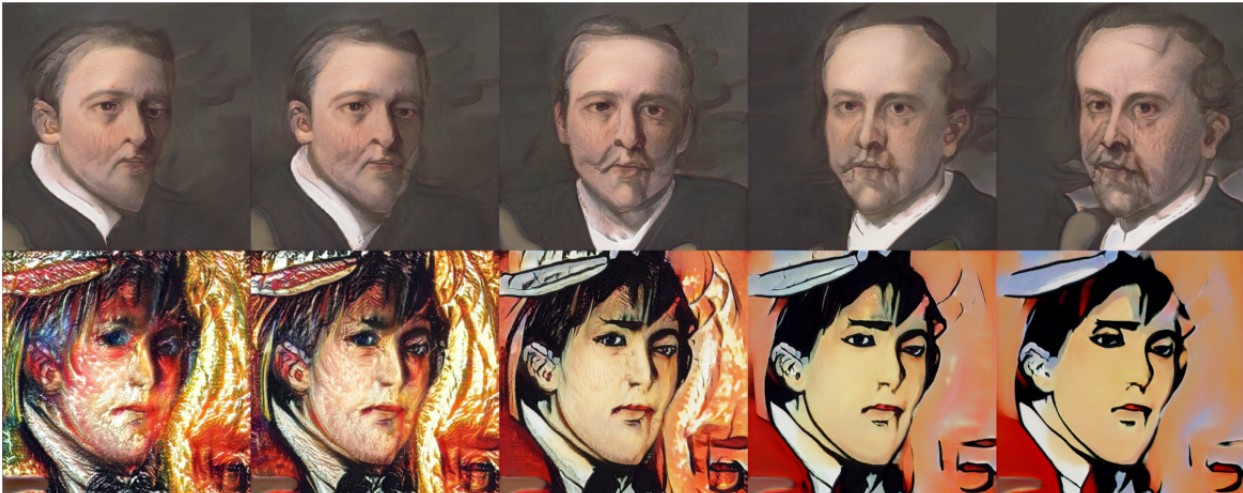

(b) "Head Rotation" and "Simple Strokes" edits on StyleGAN Wikiart without truncation psi

Figure 6: Quality of images generated by StyleGAN before and after applying the "truncation trick".

## 5   Discussion

After performing our experiments, we feel that the results justify the claims of the paper. This is further bolstered by the fact that the proposed method worked on different datasets which were not covered by the original authors.

### 5.1   What was easy

Verifying the claims of the paper was easy as the author's code was well documented and clearly written. The paper was well organized and provided a lot of examples on various datasets to demonstrate exactly how their algorithm works. The authors ensured that all the figures in the paper had accompanying code to recreate them,

NVIDIA's implementation of StyleGAN and StyleGAN2 provided access to well written API's which we could integrate easily into the author's codebase. We did not have to create our own wrappers by accessing the weights of the pretrained models.

## 5.2 What was difficult

While running out experiments, we noticed that the there was a small difference in the RGB values of the recreated images. This was due to the difference in the random values generated by PyTorch-CPU, PyTorch-GPU and Numpy random number generators even when seeded with the same seed. The noise variables in the StyleGAN networks were not identical because of this. This resulted in minute differences in background artifacts of the images.

We were not able to replicate the author's experiments on BigGAN-512 deep due to time constraints.

## 5.3 Communication with original authors

While conducting our experiments, we did not contact the original authors. The paper and codebase were organized well and aided us in effectively reproducing and validating the authors' claims.

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
