# OpenReview forum: "Reproduction of GANSpace"
_ML_Reproducibility_Challenge/2021/Fall — RC2021_

### Official Review · Reviewer_f6h7 · 2022-02-20
**Reproduction of GANSpace**

**Rating:** 9
**Confidence:** 5

**Review:**

The authors reimplemented the code in TensorFlow. The authors reproduced the results and verify the claims made by the authors for the StyleGAN and StyleGAN2. The results are interesting. The manuscript is written very nicely.

---

### Official Review · Reviewer_qmsS · 2022-03-01
**The paper is well written and the contributions are enough to get accepted!**

**Rating:** 7
**Confidence:** 4

**Review:**

Thanks for your great paper!

Summary: The paper managed to reimplement the original PyTorch code in Tensorflow 1.x and successfully verified the defined 3 claims that PCA could be used to guide important directions in the GAN's latent space and the GAN's output could be controlled at different levels in an unsupervised manner.

Pros:

- The authors used consistent mathematical notations as the original paper
- The structures of the paper are really clear and it is very easy to conclude the authors' contributions
- The authors posed some new edits in Section 4:
    - tried the method on the Stylegan2 Beetles dataset
    - found edits transferable across similar datasets
    - found enabling truncation while applying edits on StyleGAN images improved their quality as well

Cons:

- There are some small spelling errors
For example, 4. results: orignal -> original
- It would be better if the authors could add more explanations to the results section, like how to define image quality?
- From Figure 5 (b), we can see that the "hair color edit" also changes the face in the image, which is not the same as Figure 5 (a), does the statement that edits are transferable strictly hold?

---

### Official Review · Reviewer_ypr9 · 2022-03-09
**Reasonable reproduction of paper results**

**Rating:** 6
**Confidence:** 4

**Review:**

This review attempts to reproduce and add additional insight to the GANSpace paper by NVIDIA authors. The review carries out the following:

- Explain the workings of GANSpace, with explanation of the machinery behind it on how the latent space is decomposed through PCA. The distinction between the StyleGAN2 setup and BigGAN is noted. However, it is not explained in great detail.
- Reproduce the experiments in the paper (code is provided):
* Supervised vs unsupervised edits, vary PCA vs random directitons (show consistency of style and pose)
* Additional experiments not in paper: Beetles dataset and transfer edits between anime and FFHQ, effect of 'truncation trick' on wikiart dataset

Overall, it seems to be a reasonable attempt to reproduce the paper results. It seems to be lacking somewhat in insight though - I would have hoped that the authors had attempted to explain results through the lens of the PCA analyses a bit more.

---

### Meta-Review · Program_Chairs · 2022-04-09

**Recommendation:** Accept
**Confidence:** 5

**Metareview:**

The paper would additionally benefit from additional explanations, PCA analysis, and experiments expanding Figure 5.  The paper is accepted.

---

### Decision · Program_Chairs · 2022-04-09

**Decision:**

Accept

**Comment:**

Following the recommendation of reviewers and meta-reviewer, the paper is accepted for ML Reproducibility Challenge 2021, and will be published in the upcoming special edition of ReScience Journal.